# Evaluating Dataset Watermarking for Fine-tuning Traceability of Customized Diffusion Models: A Comprehensive Benchmark and Removal Approach

## Abstract

Recently, numerous fine-tuning techniques for diffusion models have been developed, enabling diffusion models to generate content that closely resembles a specific image set, such as specific facial identities and artistic styles. However, this advancement also poses potential security risks. The primary risk comes from copyright violations due to using public domain images without authorization to fine-tune diffusion models. Furthermore, if such models generate harmful content linked to the source images, tracing the origin of the fine-tuning data is crucial to clarify responsibility. To achieve fine-tuning traceability of customized diffusion models, dataset watermarking for diffusion model has been proposed, involving embedding imperceptible watermarks into images that require traceability. Notably, even after using the watermarked images to fine-tune diffusion models, the watermarks remain detectable in the generated outputs. However, existing dataset watermarking approaches lack a unified framework for performance evaluation, thereby limiting their effectiveness in practical scenarios. To address this gap, this paper first establishes a generalized threat model and subsequently introduces a comprehensive framework for evaluating dataset watermarking methods, comprising three dimensions: **Universality**, **Transmissibility**, and **Robustness**. Our evaluation results demonstrate that existing methods exhibit universality across diverse fine-tuning approaches and tasks, as well as transmissibility even when only a small proportion of watermarked images is used. In terms of robustness, existing methods show good performance against common image proces sing operations, but this does not match real-world threat scenarios. To address this issue, this paper proposes a practical watermark removal method that can completely remove dataset watermarks without affecting fine-tuning, revealing their vulnerabilities and pointing to a new challenge for future research.

## 1 Introduction

Currently, diffusion models have been widely adopted for customized content generation via fine-tuning techniques. These methods enable the adaptation of models to specific datasets, such as the individual identity and image style, As shown in figure 1.While these fine-tuning methods significantly enhances the utility of diffusion models, it also raises serious security and ethical concerns. A primary issue is the risk of copyright infringement, particularly when proprietary images are utilized for fine-tuning without authorization. Furthermore, the generation of inappropriate or harmful content by fine-tuned models requires the implementation of source traceability mechanisms to establish clear accountability. To address this challenge, dataset watermarking technology has been proposed to trace the outputs of diffusion models that have been fine-tuned using watermarked datasets. This technique involves embedding imperceptible watermarks into the images within a dataset intended for fine-tuning diffusion models. Notably, such watermarks are capable of persisting in the model's outputs following fine-tuning, thus facilitating post-hoc attribution. However, existing approaches exhibit varying definitions of threat models for dataset watermarking, which complicates the uniform evaluation of their performance in practical applications. Therefore, it is imperative to establish a unified evaluation framework to assess the performance of existing dataset watermarking techniques,

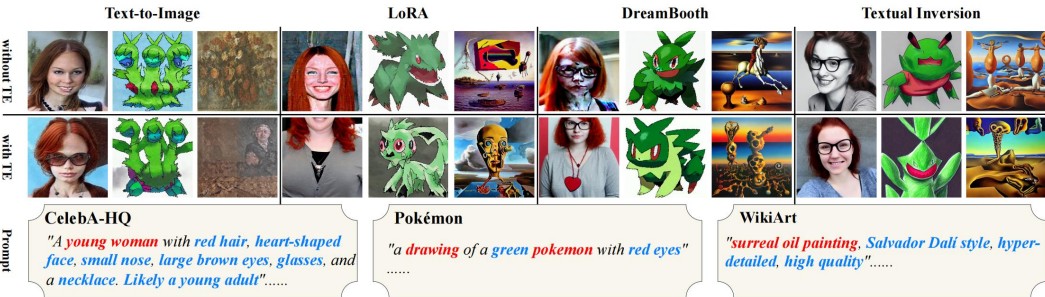

Figure 1: Visualization results of the four fine-tuning methods on three datasets. The first column shows the result without training the text encoder, the second column shows the result of training the text encoder, and the third column generates the corresponding prompt for each sample.

thereby enabling the identification of currently optimal approaches and fostering the development of more practical watermarking solutions.

To address this limitation, this paper first propose the definition of a universal threat model that characterizes the realistic adversarial scenarios encountered by dataset watermarking approaches tailored to diffusion models. Based on this threat model, we establish an evaluation framework encompassing three key dimensions: **Universality**, which refers to the applicability of dataset approaches across diverse generation tasks and various fine-tuning methods; **Transmissibility**, defined as the capability whereby the inclusion of a watermarked subset in the fine-tuning dataset is sufficient to propagate the watermarking effect, thereby ensuring the presence of watermarks in the outputs generated following fine-tuning; and **Robustness**, measuring the resilience against post-processing operations. Subsequently, we establish a comprehensive benchmark to systematically evaluate the existing dataset watermarking methods for diffusion models in terms of universality, transmissibility, and robustness. The experimental results demonstrates that existing dataset watermarking methods perform well in terms of universality and transmissibility. In robustness assessment, we initially evaluate the resilience of existing methods against common image processing operations, including noise addition, blurring, and compression. The results indicate that these methods exhibit a high level of robustness with respect to such common distortions. However, in practical scenarios, adversaries may utilize more advanced image editing techniques to remove watermarks, a challenge that has not been sufficiently addressed in prior research. To further demonstrate the significance of this issue, we propose a practical watermark removal approach that effectively removes dataset watermarks while preserving the performance of the fine-tuned model. The experimental results demonstrate that existing methods exhibit vulnerability to the proposed watermark removal approach, thereby highlighting the need for future dataset watermarking techniques to improve robustness against customized watermark removal methods. Our contributions can be summarized as follows:

- This paper defines a universal threat model for the dataset watermarking techniques that are tailored to trace diffusion models fine-tuning. Based on this, a unified evaluation framework including universality, transmissibility, and robustness is proposed to systematically assess existing methods.

- This paper establishes a comprehensive benchmark for existing dataset watermarking methods based on the evaluation framework. Experiments across diverse generation tasks and various fine-tuning methods reveal the universality and transmissibility of existing methods.

- This paper proposes a practical watermark removal method to evaluate the robustness of existing dataset watermarking techniques. Experimental results show that current dataset watermarking methods are resilient to common image processing operations but vulnerable to targeted removal attacks.

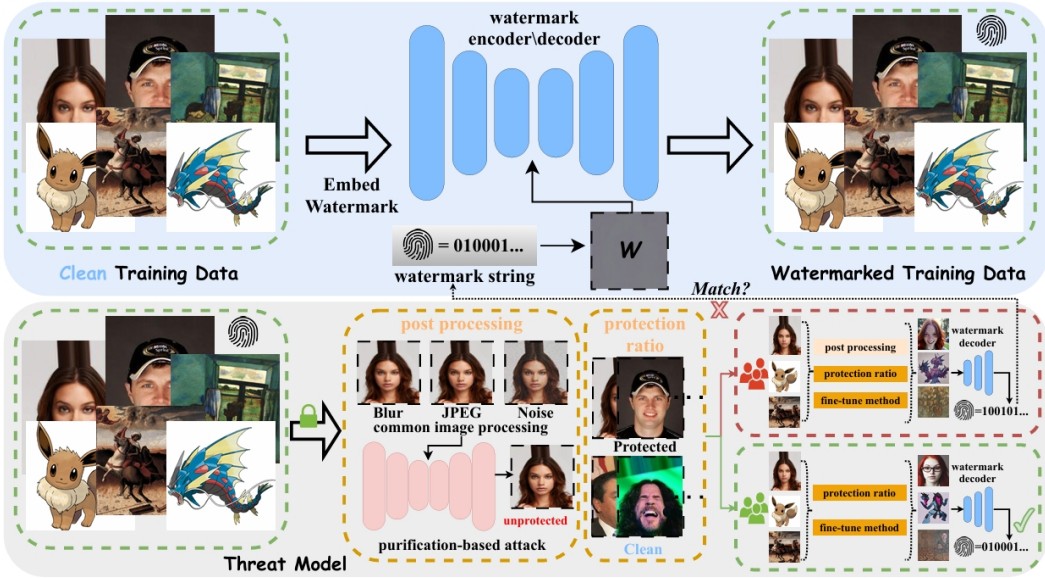

Figure 2: Overview of the threat model. **Image Owners** embed binary watermarks into datasets to establish ownership and ensure traceability. Upon acquiring the data, **Image Users** may generate customized images using various fine-tuning or model adaptation techniques. If the original watermark is successfully detected in the generated images, the protection mechanism is deemed effective; otherwise, it is considered to have failed.

## 2 RELATED WORKS

### 2.1 FINE-TUNING STABLE DIFFUSION

Due to the high computational cost of training stable diffusion models from scratch, recent research has focused on fine-tuning pre-trained models to add specific concepts. This approach leverages existing generative capabilities while greatly reducing training costs. Several fine-tuning methods have been proposed, such as Textual Inversion Gal et al. (2022), DreamBooth Ruiz et al. (2023), Custom Diffusion Kumari et al. (2023), Low-Rank Adaptation (LoRA) Hu et al. (2022), and Singular Value Diffusion (Svdiff) Han et al. (2023). These methods adapt pre-trained models in different ways to effectively introduce new concepts or styles. For instance, Textual Inversion only changes text embeddings, DreamBooth modifies the UNet architecture, Custom Diffusion targets cross-attention mechanisms, LoRA uses a low-rank matrix for parameter updates, and Svdiff adjusts singular values to create a compact parameter space.

### 2.2 IMAGE WATERMARKING

Image watermarking refers to the process of embedding imperceptible information into carrier images, primarily for the purpose of asserting and verifying copyright ownership. Traditional watermarking techniques are typically classified into spatial domain and frequency domain methods Cox et al. (2002); Navas et al. (2008); Shih & Wu (2003), where watermark data is embedded by modifying pixel intensities Cox et al. (2002), frequency coefficients Navas et al. (2008), or a combination of both Shih & Wu (2003); Kumar (2019). In recent years, an increasing number of digital watermarking approaches based on Deep Neural Networks (DNNs) Zhu et al. (2018); Zhang et al. (2019); Weng et al. (2019); Tancik et al. (2020) have been proposed, providing improved robustness and adaptability. Concurrently, several models have been developed to protect data copyrights from potential infringement by Generative Diffusion Models (GDMs). These techniques Wang et al. (2024); Cui et al. (2025b); Zhao et al. (2023); Zhu et al. (2024); Li et al. (2025); Yu et al. (2021); Cui et al. (2025a) enable traceability of unauthorized data usage through the embedding of authorized encoding information or the application of image transformation strategies.

# 3 EVALUATION FRAMEWORK

## 3.1 THREAT MODEL

This section defines a universal threat model addressing copyright protection and traceability of generated images in the context of fine-tuning diffusion models, as shown in figure 2. We define two key parties involved: **(1) Image Owner** and **(2) Image User**. The specific objectives of each party are outlined as follows:

**Image Owner:** Image Owner holds the copyright for an image dataset that may be utilized by Image User to fine-tune a diffusion model. For the purpose of copyright protection, Image Owner aims to ensure that the output of the fine-tuned model remain copyright information. Furthermore, in scenarios where the generated content is considered inappropriate, traceability to the training dataset utilized during fine-tuning should be implemented to support accountability. Therefore, Image Owner employs dataset watermarking techniques to achieve copyright protection and traceability. The embedded watermarks should satisfy the imperceptibility requirement, ensuring that it remains undetectable to human eyes and does not interfere with the generative performance of the fine-tuned diffusion model.

**Image User:** Image User collects multiple images related to the same character or style and subsequently fine-tune a diffusion model using this dataset. In cases where the fine-tuning dataset is protected through dataset watermarking, the outputs generated by the fine-tuned model should retain the embedded watermark information. On the Image User side, dataset watermarking faces three primary challenges: (1) Image User may apply various fine-tuning methods to address diverse generation tasks, with both the specific fine-tuning methods and tasks being unknown to Image Owner; (2) Image User may employ a mixed dataset containing both watermarked and original images for fine-tuning; and (3) malicious Image User may apply post-processing techniques to remove dataset watermarks prior to fine-tuning in an attempt to infringe copyrights. Therefore, dataset watermarking techniques must exhibit effectiveness across all three aforementioned challenging scenarios.

## 3.2 DIMENSIONS OF EVALUATION

Building upon the above analysis of the threat model, we categorize the requirements of dataset watermarking in diffusion models into three key dimensions, which collectively constitute the evaluation framework: **(1) Universality:** the watermarking methods should be adaptive to various fine-tuning approaches and diverse generation tasks; **(2) Transmissibility:** the watermarking methods should be capable of preserving and propagating the watermarking effect, even when only a portion of the images in the entire fine-tuning dataset are watermarked; **(3) Robustness:** the watermarking methods should be resilience against post-processing operations including both common image quality degradation and tailored watermark removal attack. The following sections will present a comprehensive evaluation of existing dataset watermarking techniques for diffusion model, based on these three dimensions.

# 4 COMPREHENSIVE BENCHMARK

This section presents a comprehensive benchmark that is established for dataset watermarking techniques within the context of tracking diffusion model fine-tuning.

## 4.1 EXPERIMENTAL SETTINGS

We evaluate four state-of-the-art dataset watermarking methods which are open source: DIAGNOSIS Wang et al. (2024), DiffusionShield Cui et al. (2025b), SIREN Li et al. (2025), and WatermarkDM Zhao et al. (2023). In experiments, all fine-tuning methods are based on stable diffusion 1.4 (SD1.4)Rombach et al. (2022). To evaluate the universality across various generation tasks, we select three datasets in distinct styles for fine-tuning: CelebA-HQ Liu et al. (2015), Pokemon Pinkney (2022), and WikiArt Wikiart (2016). To quantify the performance of these methods, we employ three evaluation metrics: FID Heusel et al. (2017), CLIP similarity Wang et al. (2023), and watermarking detection accuracy (referred to as Acc in the tables). All experiments are conducted on 4 A800 GPUs.

| Fine-Tuning Method | | Trainable Layers | | | |
|---|---|---|---|---|---|
| | | FULL-UNet | VAE | CA | TE |
| Text-to-Image | w te | ✓ | ✓ | ✗ | ✓ |
| | w/o te | ✓ | ✗ | ✓ | ✗ |
| LoRA | w te | ✗ | ✗ | ✓ | ✓ |
| | w/o te | ✗ | ✗ | ✓ | ✗ |
| DreamBooth | w te | ✓ | ✗ | ✓ | ✓ |
| | w/o te | ✓ | ✗ | ✓ | ✗ |
| Textual Inversion | w te | ✗ | ✗ | ✗ | ✓ |
| | w/o te | ✗ | ✗ | ✗ | ✗ |

Table 1: Specific configurations for the four fine-tuning methods. In UNet, "CA" denotes the Cross-Attention, and "TE" (te) refers to the Text Encoder. Among the four fine-tuning approaches, the text encoder can be configured in either a trainable mode (w/ te) or a frozen mode (w/o te).


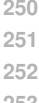

| Dataset | te | FT Metrics | Text-to-Image CLIP-T↑ | FID↓ | Acc.(%)↑ | LoRA CLIP-T↑ | FID↓ | Acc.(%)↑ | DreamBooth CLIP-T↑ | FID↓ | Acc.(%)↑ | Textual Inversion CLIP-T↑ | FID↓ | Acc.(%)↑ |
|---|---|---|---|---|---|---|---|---|---|---|---|---|---|---|
| CelebA-HQ | w/o(without) | Clean | 0.2309 | 224.70 | N/A | 0.2625 | 221.10 | N/A | 0.2565 | 207.94 | N/A | 0.2630 | 222.49 | N/A |
| | | DIAGNOSIS | 0.1947 | 247.45 | 64.00 | 0.2038 | 226.11 | 40.00 | 0.2384 | 261.38 | 20.00 | 0.2304 | 280.24 | 36.00 |
| | | DiffusionShield | 0.2184 | 259.99 | 99.83 | 0.2603 | 276.64 | 99.00 | 0.2511 | 245.11 | 100.00 | 0.2390 | 264.55 | 98.78 |
| | | WatermarkDM | 0.1814 | 286.58 | 98.44 | 0.2328 | 239.68 | 96.88 | 0.2600 | 246.40 | 95.31 | 0.2611 | 285.06 | 96.88 |
| | | SIREN | 0.2155 | 255.96 | 55.25 | 0.2593 | 274.40 | 55.04 | 0.2552 | 260.58 | 55.83 | 0.2636 | 270.83 | 53.83 |
| | w(with) | Clean | 0.2323 | 213.85 | N/A | 0.2459 | 264.44 | N/A | 0.2684 | 229.83 | N/A | 0.2649 | 226.88 | N/A |
| | | DIAGNOSIS | 0.1946 | 253.52 | 92.00 | 0.2188 | 274.43 | 46.00 | 0.2634 | 257.45 | 84.00 | 0.2311 | 274.67 | 50.00 |
| | | DiffusionShield | 0.2023 | 251.19 | 100.00 | 0.2612 | 283.58 | 100.00 | 0.2647 | 264.84 | 100.00 | 0.2358 | 271.05 | 100.00 |
| | | WatermarkDM | 0.1819 | 307.61 | 95.31 | 0.2350 | 239.29 | 93.75 | 0.2692 | 252.46 | 90.63 | 0.2600 | 280.33 | 92.19 |
| | | SIREN | 0.2174 | 256.78 | 54.63 | 0.2576 | 270.40 | 54.13 | 0.2742 | 271.23 | 54.58 | 0.2629 | 274.55 | 54.04 |
| Pokémon | w/o(without) | Clean | 0.2281 | 127.07 | N/A | 0.2856 | 158.83 | N/A | 0.2931 | 218.68 | N/A | 0.2864 | 184.81 | N/A |
| | | DIAGNOSIS | 0.2171 | 303.99 | 84.00 | 0.2597 | 277.88 | 94.00 | 0.2922 | 273.28 | 70.00 | 0.2860 | 259.04 | 70.00 |
| | | DiffusionShield | 0.2013 | 206.85 | 96.19 | 0.2843 | 247.60 | 98.78 | 0.2951 | 223.50 | 98.34 | 0.2825 | 267.79 | 98.81 |
| | | WatermarkDM | 0.2050 | 258.19 | 100.00 | 0.2447 | 191.65 | 98.44 | 0.2932 | 231.38 | 98.44 | 0.2924 | 260.91 | 95.31 |
| | | SIREN | 0.2098 | 201.16 | 57.04 | 0.2926 | 235.50 | 57.04 | 0.2953 | 211.87 | 57.08 | 0.2907 | 267.77 | 55.96 |
| | w(with) | Clean | 0.2060 | 136.15 | N/A | 0.2592 | 212.30 | N/A | 0.2902 | 200.60 | N/A | 0.2859 | 179.20 | N/A |
| | | DIAGNOSIS | 0.2050 | 338.90 | 98.00 | 0.2597 | 262.30 | 96.00 | 0.2916 | 270.58 | 44.00 | 0.2823 | 264.96 | 44.00 |
| | | DiffusionShield | 0.2139 | 220.54 | 99.64 | 0.2899 | 234.32 | 98.92 | 0.2929 | 213.00 | 99.91 | 0.2844 | 268.66 | 99.87 |
| | | WatermarkDM | 0.1943 | 273.59 | 96.88 | 0.2283 | 204.78 | 98.44 | 0.2913 | 230.89 | 90.63 | 0.2938 | 273.69 | 87.50 |
| | | SIREN | 0.2131 | 205.86 | 56.88 | 0.2839 | 244.35 | 56.58 | 0.2868 | 223.12 | 57.75 | 0.2864 | 247.05 | 56.00 |
| WikiArt | w/o(without) | Clean | 0.1401 | 320.86 | N/A | 0.2703 | 312.60 | N/A | 0.2670 | 320.12 | N/A | 0.2735 | 320.63 | N/A |
| | | DIAGNOSIS | 0.1458 | 365.44 | 90.00 | 0.2678 | 308.55 | 82.00 | 0.2680 | 310.55 | 30.00 | 0.2765 | 318.46 | 66.00 |
| | | DiffusionShield | 0.1671 | 326.28 | 100.00 | 0.2742 | 307.60 | 97.73 | 0.2755 | 318.46 | 99.91 | 0.2770 | 312.09 | 98.77 |
| | | WatermarkDM | 0.1388 | 362.88 | 90.56 | 0.1624 | 352.12 | 71.34 | 0.2480 | 336.02 | 51.31 | 0.2791 | 324.83 | 91.47 |
| | | SIREN | 0.1574 | 329.96 | 52.08 | 0.2713 | 325.00 | 53.46 | 0.2662 | 319.62 | 53.54 | 0.2789 | 329.95 | 54.54 |
| | w(with) | Clean | 0.1482 | 317.17 | N/A | 0.2703 | 306.54 | N/A | 0.2720 | 325.00 | N/A | 0.2745 | 315.48 | N/A |
| | | DIAGNOSIS | 0.1164 | 384.54 | 16.00 | 0.2612 | 301.19 | 44.00 | 0.2690 | 316.08 | 74.00 | 0.2785 | 324.58 | 30.00 |
| | | DiffusionShield | 0.1782 | 294.04 | 100.00 | 0.2657 | 306.48 | 99.80 | 0.2751 | 319.39 | 97.41 | 0.2787 | 326.07 | 99.77 |
| | | WatermarkDM | 0.1410 | 361.31 | 73.94 | 0.1659 | 351.42 | 68.00 | 0.2606 | 333.59 | 51.47 | 0.2777 | 326.14 | 49.84 |
| | | SIREN | 0.1528 | 306.82 | 51.63 | 0.2704 | 320.81 | 53.63 | 0.2742 | 329.37 | 53.71 | 0.2821 | 321.96 | 53.67 |

Table 2: The results of different watermark protection methods using various fine-tuning methods on CelebA-HQ, Pokémon and WikiArt datasets. The reference, best, and worst performance are marked by **bold**, red, and blue, respectively.

## 4.2 UNIVERSALITY EVALUATION

To evaluate the cross-method universality, we conduct assessments of these dataset watermarking techniques under various fine-tuning methodologies. Specifically, four fine-tuning methods with distinct configurations are utilized in the experiments, as detailed in Table 1. The table summarizes the trainable modules associated with each fine-tuning approach, where "te" denotes whether the text encoder is frozen during the fine-tuning process. All methods employ the default hyperparameter configurations specified in original papers, ensuring convergence of the fine-tuning process to an optimal state. Quantitative results obtained from evaluations on three datasets using four fine-tuning approaches are presented in Table 2. Regarding the generation quality after fine-tuning, fine-tuning with watermarked images adversely affects performance, potentially leading to a decrease in FID score and an increase in CLIP-T score. Specifically, the use of watermarked images for fine-tuning resulted in the most significant decline in generation performance on the Pokémon dataset.

To ensure a fair comparison of each watermarking method's ability to detect watermarks from generated images, we calibrate the watermark embedding strength across all methods, thereby guaran-

| Protection Ratio | te | FT | Text-to-Image | | | LoRA | | | DreamBooth | | | Textual Inversion | | |
|---|---|---|---|---|---|---|---|---|---|---|---|---|---|---|
| | | Metrics | CLIP-T↑ | FID↓ | Acc.(%)↑ | CLIP-T↑ | FID↓ | Acc.(%)↑ | CLIP-T↑ | FID↓ | Acc.(%)↑ | CLIP-T↑ | FID↓ | Acc.(%)↑ |
| 20% | w/o(without) | DIAGNOSIS | 0.2047 | 259.62 | 80.00 | 0.2601 | 279.06 | 40.00 | 0.2632 | 255.46 | 16.00 | 0.2634 | 283.63 | 44.00 |
| | | DiffusionShield | 0.2049 | 252.40 | 100.00 | 0.2585 | 278.47 | 100.00 | 0.2592 | 252.05 | 99.56 | 0.2672 | 289.95 | 99.91 |
| | | WatermarkDM | 0.2191 | 257.56 | 98.44 | 0.2606 | 276.72 | 89.06 | 0.2540 | 249.44 | 85.94 | 0.2618 | 285.29 | 92.19 |
| | | SIREN | 0.2145 | 256.13 | 57.25 | 0.2663 | 283.66 | 53.92 | 0.2607 | 272.31 | 55.13 | 0.2668 | 280.46 | 55.17 |
| | w(with) | DIAGNOSIS | 0.2119 | 251.84 | 76.00 | 0.2615 | 279.03 | 26.00 | 0.2630 | 265.31 | 46.00 | 0.2674 | 275.48 | 36.00 |
| | | DiffusionShield | 0.1987 | 257.69 | 100.00 | 0.2586 | 273.14 | 98.88 | 0.2697 | 277.35 | 100.00 | 0.2628 | 277.96 | 100.00 |
| | | WatermarkDM | 0.2120 | 268.10 | 96.88 | 0.2530 | 282.16 | 95.31 | 0.2721 | 260.63 | 95.31 | 0.2661 | 267.53 | 93.75 |
| | | SIREN | 0.2025 | 261.54 | 57.58 | 0.2701 | 284.60 | 53.67 | 0.2737 | 272.81 | 56.13 | 0.2634 | 279.70 | 54.21 |
| 40% | w/o(without) | DIAGNOSIS | 0.2025 | 261.50 | 44.00 | 0.2683 | 287.50 | 36.00 | 0.2598 | 247.40 | 6.00 | 0.2605 | 275.94 | 40.00 |
| | | DiffusionShield | 0.2048 | 256.56 | 100.00 | 0.2565 | 276.34 | 99.00 | 0.2534 | 259.52 | 99.00 | 0.2600 | 271.86 | 100.00 |
| | | WatermarkDM | 0.1870 | 253.48 | 89.06 | 0.2602 | 279.92 | 78.13 | 0.2532 | 252.08 | 98.44 | 0.2657 | 278.50 | 90.63 |
| | | SIREN | 0.2024 | 264.39 | 56.58 | 0.2658 | 281.70 | 54.29 | 0.2613 | 262.93 | 54.04 | 0.2671 | 279.16 | 54.00 |
| | w(with) | DIAGNOSIS | 0.2101 | 247.40 | 80.00 | 0.2646 | 275.39 | 36.00 | 0.2643 | 274.05 | 52.00 | 0.2616 | 272.49 | 78.00 |
| | | DiffusionShield | 0.2085 | 257.46 | 100.00 | 0.2645 | 276.76 | 100.00 | 0.2682 | 266.83 | 100.00 | 0.2644 | 275.95 | 98.95 |
| | | WatermarkDM | 0.1903 | 255.94 | 98.44 | 0.2570 | 278.65 | 85.94 | 0.2729 | 265.11 | 96.88 | 0.2582 | 275.12 | 95.31 |
| | | SIREN | 0.1966 | 262.58 | 55.29 | 0.2712 | 275.09 | 55.29 | 0.2692 | 281.19 | 55.25 | 0.2611 | 268.52 | 54.58 |
| 60% | w/o(without) | DIAGNOSIS | 0.2139 | 250.92 | 20.00 | 0.2599 | 273.85 | 92.00 | 0.2628 | 250.70 | 18.00 | 0.2646 | 273.28 | 52.00 |
| | | DiffusionShield | 0.2081 | 232.99 | 100.00 | 0.2573 | 280.14 | 99.73 | 0.2517 | 246.38 | 99.91 | 0.2642 | 269.59 | 99.19 |
| | | WatermarkDM | 0.2019 | 259.35 | 95.31 | 0.2613 | 274.82 | 92.19 | 0.2511 | 250.54 | 90.63 | 0.2590 | 288.34 | 89.06 |
| | | SIREN | 0.2078 | 256.26 | 53.25 | 0.2617 | 281.14 | 54.67 | 0.2673 | 257.28 | 56.33 | 0.2648 | 268.78 | 54.79 |
| | w(with) | DIAGNOSIS | 0.2066 | 246.85 | 34.00 | 0.2556 | 273.47 | 70.00 | 0.2631 | 268.76 | 26.00 | 0.2654 | 269.74 | 20.00 |
| | | DiffusionShield | 0.2064 | 241.85 | 99.94 | 0.2531 | 278.91 | 98.78 | 0.2638 | 262.69 | 100.00 | 0.2630 | 280.64 | 100.00 |
| | | WatermarkDM | 0.2002 | 269.80 | 93.75 | 0.2581 | 279.70 | 92.19 | 0.2699 | 261.96 | 93.75 | 0.2643 | 275.62 | 90.63 |
| | | SIREN | 0.2033 | 268.06 | 53.88 | 0.2587 | 280.42 | 54.33 | 0.2714 | 280.42 | 55.42 | 0.2664 | 271.40 | 54.67 |
| 80% | w/o(without) | DIAGNOSIS | 0.1962 | 268.29 | 60.00 | 0.2528 | 280.73 | 50.00 | 0.2667 | 246.12 | 70.00 | 0.2647 | 275.99 | 50.00 |
| | | DiffusionShield | 0.2126 | 251.40 | 100.00 | 0.2543 | 276.66 | 99.97 | 0.2513 | 249.16 | 99.19 | 0.2632 | 279.37 | 100.00 |
| | | WatermarkDM | 0.1969 | 257.88 | 98.44 | 0.2577 | 273.84 | 100.00 | 0.2594 | 246.49 | 98.99 | 0.2637 | 272.91 | 98.44 |
| | | SIREN | 0.2024 | 248.74 | 54.63 | 0.2696 | 282.74 | 54.13 | 0.2598 | 255.20 | 56.67 | 0.2632 | 276.49 | 54.08 |
| | w(with) | DIAGNOSIS | 0.1967 | 268.98 | 52.00 | 0.2581 | 281.58 | 70.00 | 0.2657 | 269.38 | 60.00 | 0.2600 | 280.34 | 50.00 |
| | | DiffusionShield | 0.1971 | 265.04 | 99.80 | 0.2565 | 286.48 | 100.00 | 0.2649 | 264.26 | 100.00 | 0.2626 | 280.84 | 99.00 |
| | | WatermarkDM | 0.1926 | 266.00 | 98.44 | 0.2664 | 285.49 | 98.44 | 0.2625 | 260.70 | 96.88 | 0.2696 | 267.71 | 95.31 |
| | | SIREN | 0.1996 | 264.95 | 54.13 | 0.2709 | 278.61 | 54.08 | 0.2727 | 267.00 | 55.96 | 0.2662 | 272.22 | 54.67 |

Table 3: Summary of watermark protection ratio results using different fine-tuning methods on the CelebA-HQ dataset. The best and worst performance are marked by red, and blue, respectively.

teeing that the fine-tuned generation results based on each watermarked dataset exhibit comparable performance in terms of FID and CLIP-T metrics. The detection accuracy results of watermark extraction are presented in Table 2. The experimental results demonstrate that DiffusionShield achieves the best performance, exhibiting a watermark detection accuracy approaching 100% across various datasets and fine-tuning methods. Conversely, SIREN exhibits the lowest performance, with a detection accuracy of approximately 50% across all datasets and fine-tuning methods, which is equivalent to random prediction. The poor performance of SIREN can be attributed to its original design, which is tailored for simple datasets such as CIFAR, and its limited generalization capability for more complex datasets. Consequently, SIREN struggles to effectively trace the origin data of generated images, such as portraits and artworks, in real-world applications. The remaining two methods exhibit comparatively favorable performance under specific experimental conditions. Specifically, WatermarkDM attains a watermark detection accuracy exceeding 90% on the WikiArt dataset for both Text-to-Image and Textual Inversion fine-tuning, suggesting a degree of adaptability of the watermark to varying text conditions. Nevertheless, the detection accuracy of this method declines substantially when fine-tuning is applied to the text encoder. DIAGNOSIS presents satisfactory performance across four fine-tuning methods on the Pokémon dataset. The adoption of fine-tuning the text encoder also significantly decreases the detection accuracy.

## 4.3 Transmissibility Evaluation

In real-world applications, users may employ mixed datasets comprising both unwatermarked original images and watermarked traceable images to fine-tune diffusion models. Therefore, it is essential to assess the transmissibility performance of existing dataset watermarks, specifically whether the watermarks remain intact after the model has been fine-tuned using a subset of watermarked images. To this end, we conduct mixed fine-tuning using both original and watermarked data. Specifically, the proportion of watermarked images is set at 20%, 40%, 60%, and 80% respectively. Compared to the results obtained when fine-tuning with fully watermarked images, the detection accuracy of DIAGNOSIS decreases significantly, whereas the DiffusionShield remains largely unaffected. The

| Distortion Type | te | FT Metrics | Text-to-Image | | | LoRA | | | DreamBooth | | | Textual Inversion | | |
|---|---|---|---|---|---|---|---|---|---|---|---|---|---|---|
| | | | CLIP-T↑ | FID↓ | Acc.(%)↑ | CLIP-T↑ | FID↓ | Acc.(%)↑ | CLIP-T↑ | FID↓ | Acc.(%)↑ | CLIP-T↑ | FID↓ | Acc.(%)↑ |
| **Blur** | w/o(without) | DIAGNOSIS | 0.2136 | 239.78 | 100.00 | 0.2547 | 286.35 | 64.00 | 0.2503 | 262.86 | 68.00 | 0.2604 | 267.97 | 26.00 |
| | | DiffusionShield | 0.2067 | 386.99 | 100.00 | 0.2582 | 289.61 | 100.00 | 0.2383 | 351.66 | 100.00 | 0.2660 | 279.73 | 100.00 |
| | | WatermarkDM | 0.2040 | 276.36 | 98.44 | 0.2588 | 270.90 | 89.06 | 0.2523 | 276.02 | 95.31 | 0.2617 | 279.46 | 92.19 |
| | | SIREN | 0.2246 | 246.81 | 51.71 | 0.2615 | 279.04 | 52.17 | 0.2426 | 259.72 | 52.33 | 0.2648 | 282.85 | 51.88 |
| | w(with) | DIAGNOSIS | 0.2073 | 238.28 | 92.00 | 0.2526 | 278.83 | 16.00 | 0.2504 | 271.76 | 60.00 | 0.2649 | 277.83 | 58.00 |
| | | DiffusionShield | 0.2070 | 372.00 | 100.00 | 0.2538 | 292.73 | 100.00 | 0.2493 | 321.52 | 100.00 | 0.2624 | 278.38 | 100.00 |
| | | WatermarkDM | 0.2053 | 273.67 | 98.44 | 0.2650 | 260.28 | 92.19 | 0.2607 | 261.10 | 93.75 | 0.2622 | 271.23 | 90.63 |
| | | SIREN | 0.2149 | 245.44 | 51.17 | 0.2564 | 273.18 | 52.08 | 0.2595 | 274.34 | 51.63 | 0.2662 | 270.26 | 52.04 |
| **JPEG** | w/o(without) | DIAGNOSIS | 0.2097 | 253.33 | 54.00 | 0.2513 | 278.11 | 40.00 | 0.2315 | 267.46 | 50.00 | 0.2615 | 283.02 | 90.00 |
| | | DiffusionShield | 0.2223 | 254.74 | 100.00 | 0.2582 | 279.15 | 100.00 | 0.2483 | 249.81 | 99.50 | 0.2558 | 274.64 | 99.98 |
| | | WatermarkDM | 0.2049 | 284.84 | 100.00 | 0.2618 | 268.99 | 95.31 | 0.2440 | 262.14 | 98.44 | 0.2709 | 270.01 | 93.75 |
| | | SIREN | 0.2149 | 253.97 | 52.58 | 0.2577 | 273.47 | 51.63 | 0.2591 | 254.38 | 54.42 | 0.2637 | 276.30 | 51.88 |
| | w(with) | DIAGNOSIS | 0.2088 | 257.33 | 82.00 | 0.2550 | 268.54 | 94.00 | 0.2646 | 270.64 | 76.00 | 0.2649 | 272.11 | 100.00 |
| | | DiffusionShield | 0.2194 | 277.31 | 100.00 | 0.2648 | 274.72 | 99.98 | 0.2703 | 259.33 | 99.67 | 0.2598 | 270.88 | 100.00 |
| | | WatermarkDM | 0.2128 | 276.14 | 92.19 | 0.2624 | 270.72 | 95.31 | 0.2646 | 253.28 | 90.63 | 0.2660 | 273.31 | 89.06 |
| | | SIREN | 0.2133 | 252.88 | 53.25 | 0.2591 | 272.66 | 52.21 | 0.2708 | 264.26 | 53.08 | 0.2615 | 287.28 | 52.21 |
| **Noise** | w/o(without) | DIAGNOSIS | 0.2213 | 264.12 | 60.00 | 0.2614 | 279.73 | 70.00 | 0.2485 | 261.32 | 74.00 | 0.2694 | 272.99 | 36.00 |
| | | DiffusionShield | 0.1990 | 388.39 | 82.16 | 0.2629 | 307.26 | 92.52 | 0.2400 | 369.91 | 88.67 | 0.2697 | 274.24 | 100.00 |
| | | WatermarkDM | 0.2055 | 387.67 | 98.44 | 0.2640 | 283.23 | 98.44 | 0.2514 | 323.49 | 92.19 | 0.2602 | 270.56 | 95.31 |
| | | SIREN | 0.2072 | 313.27 | 50.67 | 0.2701 | 282.57 | 51.25 | 0.2593 | 263.95 | 49.96 | 0.2627 | 279.44 | 52.38 |
| | w(with) | DIAGNOSIS | 0.2157 | 270.54 | 100.00 | 0.2613 | 278.00 | 34.00 | 0.2718 | 282.14 | 96.00 | 0.2678 | 267.09 | 100.00 |
| | | DiffusionShield | 0.2081 | 392.11 | 88.30 | 0.2500 | 303.06 | 93.08 | 0.2490 | 319.35 | 86.98 | 0.2607 | 275.41 | 98.89 |
| | | WatermarkDM | 0.2098 | 422.80 | 96.88 | 0.2670 | 274.59 | 98.44 | 0.2805 | 287.30 | 90.63 | 0.2647 | 271.09 | 90.63 |
| | | SIREN | 0.2015 | 341.88 | 49.79 | 0.2576 | 283.16 | 50.54 | 0.2737 | 276.27 | 50.08 | 0.2627 | 274.34 | 52.17 |

Table 4: Summary of natural distortion to watermark protection results under different fine-tuning methods on CelebA-HQ. The best and worst performance are marked by red, and blue, respectively.

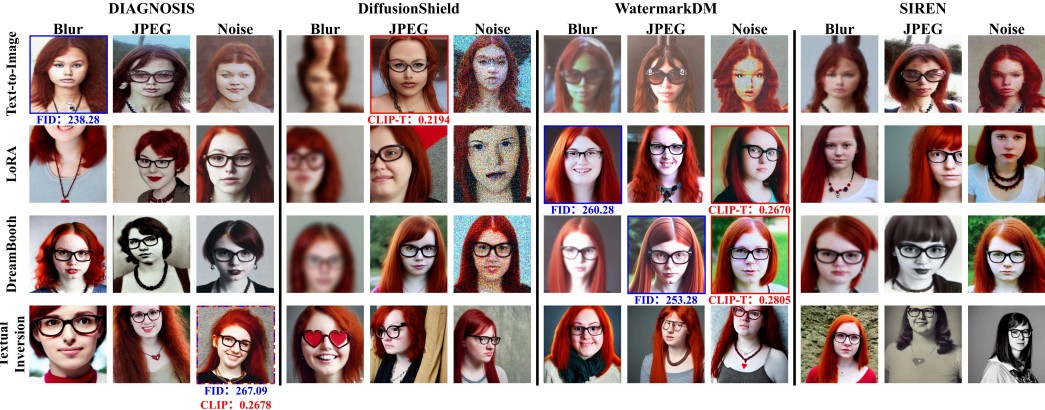

Figure 3: The visualization of generation results after applying natural distortion. The figure indicates the optimal FID score and CLIP-T similarity for each fine-tuning approach.

other two methods fail to demonstrate the traceability of the dataset watermark to the results generated after fine-tuning.

## 4.4 ROBUSTNESS EVALUATION UNDER COMMON DISTORTIONS

Dataset watermarking methods may be vulnerable to potential post-processing operations, which can compromise the integrity of the embedded watermark signals. Hence, it is crucial to evaluate the robustness of existing dataset watermarking approaches against a variety of post-processing techniques. We categorize potential post-processing operations into two classes: common image processing techniques and specifically designed watermark removal methods. The first category of post-processing operations may lead to overall quality degradation in watermarked images, consequently resulting in diminished performance during fine-tuning. We evaluate the robustness of existing methods under common image degradation operations, including Gaussian noise, Gaussian blur, and JPEG compression. These three types of image processing techniques are typically introduced during the transmission and processing of datasets, rather than being specifically applied for

Figure 4: The architecture of **DeAttack**. A unified framework for watermark removal, utilizing image degradation and restoration processes.

malicious attacks. As shown in Table 4, both DIAGNOSIS and DiffusionShield exhibit robustness in these conditions.

## 5 WATERMARK REMOVAL APPROACH

As some watermarking methods remain robust under common image degradations, tailored strategies are required for effective watermark removal. In the Regeneration Attack approaches proposed by Zhao et al.Bansal et al. (2023) for watermark removal, the destructive process is implemented by adding Gaussian noise either in the pixel space or the representation space. However, this choice may not be optimal for wa termark removal tasks. The following presents a theoretical analysis.

**Additive Gaussian noise in pixel space.** Let $I(x, y)$ denote the original image, and suppose a watermark has been embedded to produce a watermarked image $I_w(x, y) = I(x, y) + \alpha W(x, y)$, where $W(x, y)$ denotes the watermark signal and $\alpha \ll 1$ is the embedding strength. An attacker may attempt to disrupt the watermark by adding zero-mean Gaussian noise with variance $\sigma^2$:

$$I'(x, y) = I_w(x, y) + n(x, y), \quad n(x, y) \sim \mathcal{N}(0, \sigma^2) \tag{1}$$

Applying the Fourier transform yields:

$$\hat{I}'(u, v) = \hat{I}_w(u, v) + \hat{n}(u, v) \tag{2}$$

Since white Gaussian noise has a flat power spectral density, $\hat{n}(u, v)$ contributes equally to all frequency bands:

$$\mathbb{E}[|\hat{n}(u, v)|^2] = \sigma^2 MN \tag{3}$$

This uniform perturbation can effectively mask the spectral structure of $\hat{W}(u, v)$, particularly when the watermark is embedded in mid- or high-frequency regions. However, the random and unstructured nature of the noise implies that some watermark energy may remain intact, especially under robust extraction algorithms.

**Gaussian blur in pixel space.** Another attack method applies Gaussian blur to the watermarked image:

$$I'(x, y) = (I_w * G_\sigma)(x, y) \tag{4}$$

where $G_\sigma$ is a Gaussian kernel. In the frequency domain, this operation becomes:

$$\hat{I}'(u, v) = \hat{I}_w(u, v) \cdot H(u, v) \tag{5}$$

with

$$H(u, v) = \exp(-2\pi^2 \sigma^2 (u^2 + v^2)) \tag{6}$$

This exponential low-pass filter suppresses high-frequency components where watermarks are often embedded. The watermark spectrum is thus attenuated as:

$$\alpha \hat{W}(u, v) \cdot H(u, v) \to 0 \quad \text{as } \|(u, v)\| \to \infty \tag{7}$$

| Method | VAE (Bmshj2018) | | | VAE (Cheng2020) | | | Diffusion | | | SwinIR (denoise) | | | SwinIR (JPEG AR) | | | IRNeXt (deblur) | | |
|---|---|---|---|---|---|---|---|---|---|---|---|---|---|---|---|---|---|---|
| | CLIP-T↑ | FID↓ | Acc.(%)↑ | CLIP-T↑ | FID↓ | Acc.(%)↑ | CLIP-T↑ | FID↓ | Acc.(%)↑ | CLIP-T↑ | FID↓ | Acc.(%)↑ | CLIP-T↑ | FID↓ | Acc.(%)↑ | CLIP-T↑ | FID↓ | Acc.(%)↑ |
| DIAGNOSIS | 0.2594 | 273.36 | 72.00 | 0.2613 | 283.34 | 64.00 | 0.2628 | 280.06 | 84.00 | 0.2557 | 269.89 | 70.00 | 0.2531 | 275.03 | 68.00 | 0.2508 | 278.79 | 78.00 |
| DiffusionShield | 0.2611 | 268.64 | 99.99 | 0.2609 | 282.65 | 99.78 | 0.2654 | 274.22 | 100.00 | 0.2614 | 268.66 | 98.39 | 0.2565 | 272.21 | 100.00 | 0.2519 | 257.92 | 100.00 |
| WatermarkDM | 0.2571 | 270.79 | 62.50 | 0.2652 | 277.76 | 57.81 | 0.2696 | 269.35 | 54.69 | 0.2589 | 269.68 | 51.56 | 0.2579 | 268.63 | 48.44 | 0.2525 | 258.28 | 56.25 |
| SIREN | 0.2601 | 270.64 | 51.83 | 0.2593 | 285.56 | 52.29 | 0.2554 | 277.21 | 51.92 | 0.2584 | 273.65 | 52.38 | 0.2555 | 275.34 | 52.38 | 0.2519 | 263.98 | 52.21 |

Table 5: Results of different DeAttack methods on CelebA-HQ (LoRA, w/o te). The best and worst performance are marked by red, and blue, respectively.

Compared to additive noise, this strategy systematically targets and reduces the energy of frequency-domain watermarks, making it particularly effective in scenarios where the watermark is localized in the high-frequency spectrum.

**Additive Gaussian noise in latent space.** In encoder-based models, images are mapped to a latent representation $z = E(I)$, where additive Gaussian noise $\epsilon \sim \mathcal{N}(0, \sigma^2 I)$ can be injected:

$$z' = z + \epsilon \tag{8}$$

Unlike pixel-space noise, which perturbs all frequencies uniformly, latent-space noise leads to structured, model-dependent distortions in the image. Due to the encoder's compression and abstraction, high-frequency details are often underrepresented in $z$. As a result, latent noise tends to produce low-to mid-frequency artifacts, which may be less effective in suppressing high-frequency watermarks.

To establish a more general framework of regeneration attacks for watermark removal, we first formulate the process in a simplified form:

$$x = R(D(x)). \tag{9}$$

Here, D denotes a degradation process, and R denotes the image restoration process. The degradation D can be categorized into three types: additive noise in the pixel space, additive noise in the representation space, and deterministic degradation (cold degradation) in the pixel space Bansal et al. (2023). The image restoration process can be implemented using models such as DAE, VAE, diffusion models, or other restoration architectures.

As shown in Figure 4, we propose *DeAttack*, a unified framework that leverages various image degradation and restoration processes to remove watermarks. The network architecture used in DeAttack can be uniformly described as an autoencoder, with optional restoration blocks inserted between the encoder and decoder. Image degradation is applied either before the encoder in the pixel space, or after the encoder in the representation space via Gaussian noise.

To better remove the watermark, we train IRNeXt Cui et al. (2023; 2024) on the DIV2K Agustsson & Timofte (2017), Flickr2K, and WED Ma et al. (2016) datasets by generating Gaussian-blurred images with a kernel size of $71 \times 71$ and standard deviation $\sigma = 15$. In addition, we adopt two pretrained SwinIR Liang et al. (2021) models for denoising and JPEG compression artifact reduction to perform image-restoration-based watermark removal. We also evaluate the three regeneration attack methods proposed by Zhao et al. Bansal et al. (2023). As shown in Table 5, our IRNeXt-based model achieves effective watermark removal while preserving image quality.

## 6 CONCLUSION

This paper analyzes image watermarking techniques aimed at tracing unauthorized fine-tuning of data by Stable Diffusion. Experimental results show these methods lack robustness in real-world scenarios. The results of some existing methods exhibit universality across diverse fine-tuning approaches and tasks, as well as transmissibility even when only a small proportion of watermarked images is used. Finally, we propose DeAttack, a unified watermark removal framework based on image degradation and restoration. We assess how various types of noise and degradation impact watermark removal. Results show our method outperforms existing approaches under DeAttack and could inspire more robust watermarking techniques.

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

# A APPENDIX

The appendix provides additional information that complements the main text. It includes implementation details, extended experimental results, and further analyses that were excluded from the main paper due to space constraints. All experiments were conducted following the same setup and evaluation protocols described in the main manuscript.

## A.1 ETHICS STATEMENT

This study adheres to the ICLR ethical guidelines. No human subjects or animal experiments were involved in this research. All datasets used, including {CelebA, Pokémon, and WikiArt}, were obtained in accordance with the relevant usage guidelines to ensure no privacy was violated.

## A.2 REPRODUCIBILITY STATEMENT

We have made every effort to ensure the reproducibility of the results presented in this paper. The experimental setup, including the training procedures, model configurations, and hardware details, are all described in detail in the paper. In addition, all the public datasets used in this paper are publicly available, which ensures the consistency and reproducibility of the evaluation results.

## A.3 LLM USAGE

Large language models (LLMs) were only used to enhance the language quality of the manuscript, including rewriting, grammar checking, and improving fluency. They were not involved in the research conception, methodology, or experimental design. All scientific content was completed by the authors, and the use of LLMs was ensured to comply with academic norms, without any plagiarism or academic misconduct.

## A.4 DATASET DETAILS

In this section, we present the details of the three datasets used in this paper.

- **CelebA** Liu et al. (2015): This dataset comprises facial images of various celebrities, each paired with a descriptive caption generated by the LLaVA Liu et al. (2023) model. Given that the original dataset is large in size and the fine-tuning diffusion model requires a low amount of data, we sampled 1,000 images from the original dataset for experimental use.
- **Pokémon** Pinkney (2022): This dataset comprises 833 high-quality Pokémon images, each paired with a text caption generated by the BLIP Li et al. (2022) captioning model.
- **WikiArt** Wikiart (2016): This dataset containing 81444 pieces of visual art from various artists, taken from WikiArt.org, along with class labels for each image: "artist", "genre" and "style".

## A.5 50 PROMPTS FOR GENERATION

This section lists in detail the prompts used to generate samples after fine-tuning the stable diffusion model. Each dataset has 50 prompts.

### A.5.1 CELEBA

1. "A young woman with red hair, heart-shaped face, small nose, large brown eyes, glasses, and a necklace. Likely a young adult."
2. "A young bald man with a beard and round face, wearing a football helmet. He has thick lips and a large nose."
3. "A blonde young woman with a heart-shaped face, small nose, thin lips, wearing a black dress and a necklace. Smiling."
4. "A young woman with dark hair, heart-shaped face, full lips, straight nose, and a necklace. Likely in her late teens."

5. "A bald man with glasses, wide face, thick lips, wearing a suit and standing at a microphone. He is an adult."

6. "An elderly white man with glasses, wide face, large nose, thin mouth, and beard. Wearing a tie and brown jacket."

7. "A young blonde woman with glasses, small nose, wide mouth, brown eyes, a bracelet, and a nose piercing. No facial hair."

8. "A smiling young man with a round face, glasses, beard, brown eyes, and wearing a suit and tie. A flag is behind him."

9. "A man with a wide face, thick mustache, black hair, and glasses. Smiling, with no other accessories visible."

10. "A young woman with dark hair, brown cat-like eyes, heart-shaped face, wide mouth, small nose, and black dress."

11. "A man with blonde hair, wide face, small nose, thick lips, large eyes, and glasses. Wearing a white shirt and smiling."

12. "A young woman with a narrow heart-shaped face, dark hair, large eyes, small nose, thin lips, wearing pink dress and necklace."

13. "A smiling young blonde woman with heart-shaped face, brown eyes, small nose, glasses, thick eyebrows, and a necklace."

14. "A young man with glasses, dark hair, large nose, black eyes, strong jawline, and straight mouth. Likely a young adult."

15. "A woman with blonde bobbed hair, red dress, heart-shaped face, small nose, smiling. Possibly young, not clearly elderly."

16. "A young blonde woman with large expressive eyes, small nose, thin mouth, glasses, white shirt, and heart-shaped face."

17. "A smiling young blonde woman with heart-shaped face, blue eyes, glasses, small wide nose, thick lips, and jewelry."

18. "A young woman with brown eyes, glasses, thick lips, pearl necklace, heart-shaped face, small nose, and a smile."

19. "A young woman with auburn hair, glasses, brown eyes, wide mouth, heart-shaped face, and a necklace."

20. "A young woman with a ponytail, dark hair, glasses, small nose, full mouth, black clothes, and heart-shaped face."

21. "An elderly bald man with a beard, wide thick face, large nose, wearing a jacket and hat. Seated in front of a camera."

22. "A young man with shaved sides and ponytail, large brown eyes, wide upturned nose, glasses, beard, and rectangular face."

23. "A smiling young blonde woman with heart-shaped face, pointed nose, full lips, earrings, and brown eyes. No glasses."

24. "A young blonde woman with wide face, small wide nose, thick lips, large earrings, necklace, and a smile."

25. "A young bald man with glasses, beard, large nose, wide face, thick eyebrows, and a black jacket. Likely young adult."

26. "A goofy young man with round face, large eyes, small nose, thin mouth, glasses, and a playful smile."

27. "A young man with shaved head, large nose, wide open eyes, black hoodie, wide mouth. Possibly a teen or young adult."

28. "A young woman in pink dress with round face, glasses, pink bow, large brown eyes, holding a cherry in her mouth."

29. "An elderly bald man with glasses, large nose, thick mustache, red-striped shirt, tie, and a big smile."

30. "A man with long hair, beard, glasses, large nose, wide mouth, wearing a black shirt and jacket. Likely adult."

31. "A young blonde woman with heart-shaped face, large brown eyes, small nose, necklace, and a thin mouth. Wearing a dress."

32. "A smiling woman with almond-shaped eyes, wide nose, large mouth, pink dress, blonde hair. Likely a young adult."

33. "A young woman with red shirt, heart-shaped face, large dark eyes, full lips, small nose, necklace, and ponytail."

34. "A smiling young woman with heart-shaped face, small nose, large brown eyes, pink bathing suit, and pink headband."

35. "A young man in a suit and tie with round face, small nose, brown eyes, glasses, beard, and a thin mouth."

36. "A young woman with long black hair, glasses, small nose, heart-shaped face, necklace, and a thin mouth. Likely teen."

37. "A smiling young woman with heart-shaped face, wide mouth, large eyes, necklace, and ponytail. Possibly young adult."

38. "An elderly person with wide face, large black eyes, round nose, bushy eyebrows, suit, tie, and hat."

39. "A smiling young woman with curly dark hair, large brown eyes, full lips, small nose, necklace, and heart-shaped face."

40. "A man with beard, sunglasses, black suit and tie, large nose, wide mouth, and prominent chin. Handsome appearance."

41. "An elderly bald man with white beard, glasses, very wide face, small nose, thick mouth, wearing a suit and tie."

42. "A woman with round face, glasses, blonde hair, thick lips, blue shirt, small wide nose, and a friendly smile."

43. "A man with shaved head and beard, blue shirt, wide mouth, large nose, small blue eyes, and a youthful appearance."

44. "A young woman with dark straight hair, glasses, brown eyes, small nose, full lips, necklace, bracelet, and oval face."

45. "A thin-faced bald man with glasses, large nose, close-set eyes, suit and tie, looking directly at the camera."

46. "A young woman with heart-shaped face, dark hair, large expressive eyes, full mouth, small nose, and earrings."

47. "A young woman with blue hair, red dress, large round eyes, thick lips, wide face, necklace, and blue eyes."

48. "A person with long black hair, white shirt, large black eyes, wide nose, glasses, and nose piercing. Teen or adult."

49. "A smiling young blonde woman with round face, small nose, blue eyes, glasses, necklace, and red background."

50. "A young man with round face, glasses, full beard, small nose, wearing a suit and tie. Appears formally dressed."

### A.5.2 POKÉMON

1. "a drawing of a green pokemon with red eyes"

2. "a cartoon monkey flying with a bone in its mouth"

3. "a drawing of a purple dragon with spikes on it's head"

4. "a drawing of a cat sitting on top of a flower"

5. "a pink bird with orange eyes and a pink tail"

6. "a cartoon bee with a big smile on it's face"

7. "a blue cartoon character with a target in his hand"

8. "a cartoon bird with a green leaf on its head"

9. "a drawing of a blue dinosaur with wings"

10. "a drawing of a green pokemon sitting on top of a leaf"

11. "a very cute looking pokemon type"

12. "a drawing of a shark with its mouth open"

13. "a cartoon character with a mushroom on his head"

14. "a drawing of a cat wearing a helmet"

15. "a drawing of a cat with a pink tail"

16. "a cartoon elephant with a red nose and orange ears"

17. "a drawing of a black and white animal with horns"

18. "a drawing of a purple and white animal"

19. "a drawing of a red and yellow insect"

20. "a drawing of a green and yellow lizard"

21. "a drawing of a blue and orange pokemon"

22. "a drawing of a gray and white pokemon"

23. "a cartoon picture of a green vegetable with eyes"

24. "a drawing of a green cartoon character with a sad look"

25. "a cartoon giraffe with a ball in its mouth"

26. "a cartoon bird with a hat on its head"

27. "a cartoon dog is standing in a pose"

28. "a drawing of a star with a red eye"

29. "a cartoon turtle with a tree on its back"

30. "a drawing of a pink cartoon character"

31. "a drawing of a fox with wings on it's back"

32. "a blue and yellow cartoon character with its mouth open"

33. "a cartoon mouse with a pink shirt and tie"

34. "a cartoon character with a yellow shirt and blue pants"

35. "a drawing of a fish with a horn on it's head"

36. "a drawing of a white and red pokemon"

37. "a drawing of a blue fish with yellow eyes"

38. "a cartoon bunny flying through the air"

39. "a drawing of a small animal with a pink nose"

40. "a blue and white cartoon character flying through the air"

41. "a green and yellow toy with a red nose"

42. "a drawing of a woman in a pink dress with a dragon head"

43. "a cartoon character with a magnifying glass"

44. "a drawing of a blue sea turtle holding a rock"

45. "a cartoon bear with a ring around its neck"

46. "a cartoon cat is holding onto a leash"

47. "a cartoon rat with its mouth open and it's mouth wide open"

48. "a green bird with a red tail and a black nose"

49. "a cartoon sheep is kicking a soccer ball"

50. "a close up of a cartoon character with big eyes"

### A.5.3 WikiArt

1. "surreal oil painting, Salvador Dalí style, hyper-detailed, high quality"
2. "romantic landscape, 19th-century French painting, soft brushwork, ultra high-res"
3. "cubist still life, abstract geometric shapes, Picasso-inspired, vibrant colors"
4. "impressionist river scene, vivid brush strokes, Claude Monet style, realistic lighting"
5. "art nouveau floral pattern, elegant flowing lines, Alphonse Mucha inspired, intricate details"
6. "German expressionist portrait, emotional color palette, dramatic, cinematic lighting"
7. "Russian avant-garde constructivist poster, vintage style, bold typography, clean vector"
8. "hyper-realistic Baroque portrait, dramatic chiaroscuro, Rembrandt style, 8K"
9. "abstract color field painting, Rothko inspired, vivid colors, minimalist"
10. "Italian Renaissance fresco, mythological figures, high detail, realistic faces"
11. "minimalist geometric abstraction, Malevich style, pure shapes, modern design"
12. "medieval illuminated manuscript, gold leaf, intricate patterns, ancient calligraphy"
13. "gothic cathedral interior, stained glass, atmospheric light, photorealistic"
14. "Japanese woodblock print, Hokusai style, traditional ukiyo-e, fine linework"
15. "fauvist landscape, intense color contrasts, Matisse inspired, expressive painting"
16. "surreal dreamscape, Magritte style, hyper-realistic, conceptual art"
17. "art deco poster, glamorous 1920s woman, vintage illustration, high detail"
18. "Russian symbolist painting, mystical, ethereal lighting, rich textures"
19. "rococo palace interior, pastel colors, ornate details, photorealistic"
20. "Dutch golden age still life, flowers and fruits, realistic lighting, master painting"
21. "pre-Raphaelite portrait, medieval-inspired, flowing hair, detailed textile"
22. "Chinese ink landscape, shan shui style, misty mountains, traditional painting"
23. "Bauhaus modernist architectural drawing, clean lines, geometric composition"
24. "Italian futurist cityscape, motion blur, dynamic angles, vibrant"
25. "Byzantine mosaic, religious icon, gold tesserae, intricate details"
26. "Spanish romantic painting, dramatic history scene, vivid brushwork, realistic"
27. "symbolist fantasy scene, allegorical figures, mystical atmosphere, high detail"
28. "social realism mural, workers, propaganda style, bold colors, large format"
29. "abstract expressionist painting, chaotic brushstrokes, Pollock style, large canvas"
30. "Venetian rococo carnival scene, masked figures, ornate costumes, detailed"
31. "French rococo pastoral painting, elegant people, romantic light, high realism"
32. "Russian lubok folk art, storytelling style, bright colors, naive art"
33. "Egyptian revival decorative motif, hieroglyphs, ancient style, symmetrical pattern"
34. "neoclassical sculpture study, idealized human figure, marble texture, photorealistic"
35. "academic classical painting, mythological subject, realistic anatomy, dramatic light"
36. "Neue Sachlichkeit portrait, German realism, neutral colors, intense gaze"
37. "surrealist collage, Max Ernst style, dreamlike, high-res details"
38. "pre-Columbian inspired pattern, tribal geometric symbols, earthy colors"
39. "gothic illuminated manuscript page, ornate borders, medieval style, hyper-detailed"
40. "classical Greek vase painting, heroic myth scene, terracotta style, authentic"
41. "romantic seascape, stormy sky, 19th-century painting style, high detail"
42. "Renaissance-inspired religious altarpiece, golden halos, realistic faces, dramatic"

43. "art brut, outsider art style, raw brushstrokes, expressive emotion"

44. "Victorian fairy painting, delicate wings, flower garden, high detail"

45. "expressionist cityscape, angular architecture, dramatic colors, thick brush strokes"

46. "post-impressionist village scene, vivid colors, Van Gogh style, swirling strokes"

47. "orientalist painting, Middle Eastern architecture, rich textures, historical"

48. "pop art reinterpretation, classical sculpture, bright bold colors, high contrast"

49. "suprematist non-objective composition, simple shapes, modernist, clean vector"

50. "primitivist figure painting, tribal inspiration, earthy colors, simplified forms"

## A.6  EXPERIMENTAL SETTINGS

Our experiments were implemented in Python 3.10 and PyTorch 2.7.1. All experiments were performed on Ubuntu 20.04 equipped with 4 A800 GPUs.

### A.6.1  MODELS&DATASETS

We selected DIAGNOSIS Wang et al. (2024), DiffusionShield Cui et al. (2025b), SIREN Li et al. (2025), and WatermarkDM Zhao et al. (2023) as the primary watermark protection methods. These approaches safeguard the original images by embedding invisible watermarks and enable the tracing of unauthorized data usage. The evaluation primarily focuses on three protection aspects: face protection, virtual object protection, and artistic style protection, conducted on three high-resolution datasets: CelebA-HQ Liu et al. (2015), Pokémon Pinkney (2022), and WikiArt Wikiart (2016),

### A.6.2  IMPLEMENTATION DETAILS

During the watermark embedding phase, we utilized the model's default configuration and adjusted the image resolution to $512 \times 512$. In the fine-tuning phase, SD1.4 Rombach et al. (2022) was uniformly employed, and the same prompt template was applied across the four fine-tuning methods to ensure a fair comparison based on 10 images from the datasets. During the generation phase, five images were generated for each of the 50 prompts, and their average FID score and CLIP similarity were calculated.

## A.7  COMMON DISTORTION PROCESSING

We used three common distortions, which are implemented as follows:

### A.7.1  IMAGE BLUR

$$I_{\text{blur}}(x,y) = (I * G)(x,y) = \sum_{u=-k}^{k} \sum_{v=-k}^{k} I(x-u, y-v) \cdot G(u,v), \tag{10}$$

$$G(u,v) = \frac{1}{2\pi\sigma^2} \exp\left(-\frac{u^2 + v^2}{2\sigma^2}\right), \tag{11}$$

where $I$ is the original image, $G$ is a two-dimensional Gaussian kernel, $\sigma$ is standard deviation. We used a $31 \times 31$ kernel, which automatically calculates the corresponding standard deviation.

### A.7.2  JPEG COMPRESSION

For JPEG compression, we divide the image into $8 \times 8$ blocks and perform Discrete Cosine Transform (DCT), and then quantize the frequency domain coefficients $C(u,v)$, as follows:

$$C_q(u,v) = \text{round}\left(\frac{C(u,v)}{Q(u,v)}\right), \tag{12}$$

where $Q(u,v)$ is the standard quantization matrix. The distortion mainly comes from this quantization operation. The JPEG quality level used in the code is 15.

### A.7.3 GAUSSIAN NOISE

For Gaussian noise, the process is as follows:

$$I_{\text{noisy}}(x, y, c) = \text{clip}\left(I(x, y, c) + \mathcal{N}(0, \sigma^2), 0, 1\right), \tag{13}$$

where $\mathcal{N}(0, \sigma^2)$ represents Gaussian noise with mean 0 and variance $x$; the $clip$ function clamps the pixel values to the interval $[0, 1]$. The distortion is added independently in the dimension of the color channel $c$.

The overall processing flow can be described as Algorithm 1.

---

**Algorithm 1** Apply Image Distortions: Gaussian Blur, JPEG Compression, and Gaussian Noise

---

**Require:** Original image $I$; JPEG quality $q = 15$; Gaussian noise std $\sigma = 0.3$
**Ensure:** Distorted images: $I_{\text{blur}}, I_{\text{jpeg}}, I_{\text{noise}}$
 1: **Gaussian Blur:**
 2: $I_{\text{blur}} \leftarrow \text{GaussianBlur}(I, \text{kernel\_size} = 31, \sigma = 0)$
 3: **JPEG Compression:**
 4: $I_{\text{jpeg}} \leftarrow \text{JPEGEncode}(I, \text{quality} = q)$
 5: **Gaussian Noise:**
 6: $I_{\text{norm}} \leftarrow I/255$
 7: Sample $\varepsilon \sim \mathcal{N}(0, \sigma^2)$
 8: $I_{\text{noise}} \leftarrow \text{clip}(I_{\text{norm}} + \varepsilon, 0, 1)$
 9: $I_{\text{noise}} \leftarrow I_{\text{noise}} \times 255$
10: **Return:** $I_{\text{blur}}, I_{\text{jpeg}}, I_{\text{noise}}$

---

## A.8 VISUALIZATION OF DEATTACK METHOD

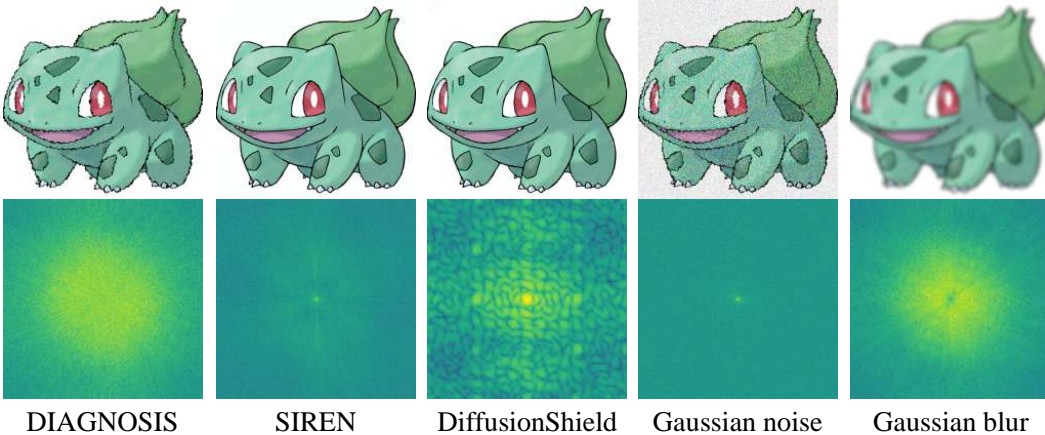

| DIAGNOSIS | SIREN | DiffusionShield | Gaussian noise | Gaussian blur |

Figure 5: Visualization of DeAttack Method.