# OpenReview forum: "Evaluating Dataset Watermarking for Fine-tuning Traceability of Customized Diffusion Models: A Comprehensive Benchmark and Removal Approach"
_ICLR.cc/2026/Conference — ICLR 2026 Conference Withdrawn Submission_

### Official Review · Reviewer_8LGY · 2025-10-25

**Soundness:** 2
**Presentation:** 2
**Contribution:** 2
**Rating:** 2
**Confidence:** 4

**Summary:**

The paper proposed a benchmark for evaluating dataset watermarking techniques used to trace fine-tuning in diffusion models. It introduced a unified evaluation framework based on three key dimensions (universality, transmissibility, and robustness) to assess the effectiveness of existing watermarking methods. Experiments show that while current methods perform well under standard conditions, they remain vulnerable to targeted watermark removal attacks. In addition, the paper proposes a watermark-removal attack, DeAttack, which leverages image degradation and restoration processes to effectively remove watermarks, exposing critical vulnerabilities and motivating stronger defenses.

**Strengths:**

1. In general, the writing of the paper is relatively clear and easy to follow.
2. The paper focuses on a meaningful and interesting topic: the systematic evaluation of current dataset watermarking techniques for copyright protection against the fine-tuning of diffusion models, and indicates three important aspects for evaluation: universality, transmissibility, and robustness.

**Weaknesses:**

1. Some discussion may not be comprehensive. For example, when applying the attack to the watermarking methods, the authors consider only applying the attack to the image used for fine-tuning. However, the attack may also be applied to the generated images after fine-tuning, or applied in both stages. In addition, there could also be a scenario where the data used for fine-tuning a diffusion model consists of watermarked data from different users, where the watermarking is done by the same method with different embedded IP messages, or by a different watermarking method.
2. The presentation of Table 5 seems to be unclear. Which regenerations in Table 5 are the baselines by Zhao et. al and Bansal et. al, and what is the meaning of VAE and diffusion in Table 5?
3. The performance of the proposed watermark removal method is limited. The performance of the proposed methods using different degradation and restoration seems to be similar to other attacking methods, where DiffusionShield still demonstrates near 100% accuracy after the attack. Moreover, the method lacks clear novelty, as the overall degradation–restoration framework closely follows prior regeneration attack designs.
4. The paper does not offer particularly novel or surprising insights into existing watermarking methods. While it emphasizes the importance of robustness against targeted attacks, this concern has already been well recognized and discussed in many previous works.
5. The evaluation in this paper only considers Stable diffusion 1.4, which limits the generality of the conclusions. It is suggested that newer versions of Stable Diffusion or alternative architectures like SDXL and DeepFloyd should be considered.
6. Small issue: the authors did not distinguish the use of \cite and \citep in in-text citations.

**Questions:**

1. In Section 3, the authors state that “in scenarios where the generated content is considered inappropriate, traceability to the training dataset utilized during fine-tuning should be implemented to support accountability.” This statement may require further clarification. If the watermark key or verification process is only known to the data owner, and that owner themselves uses the data for malicious purposes, how can accountability be established or enforced?
2. What are the numbers of fine-tuning steps considered in different fine-tuning methods? Can the early stopping of the fine-tuning influence the performance of watermark detection?
3. In Section 5, the authors mention that high-frequency details are underrepresented in the latent space of the image. Does this imply that simply compressing an image into the latent space and reconstructing it could effectively remove the watermark if the watermark is in the high-frequency domain?

---

### Official Review · Reviewer_8hQ6 · 2025-10-28

**Soundness:** 3
**Presentation:** 2
**Contribution:** 2
**Rating:** 4
**Confidence:** 5

**Summary:**

This paper systematically studies dataset watermarking for diffusion model fine-tuning and proposes a unified threat model and three-dimensional evaluation framework covering universality, transmissibility, and robustness. It also builds the first comprehensive benchmark across multiple datasets and fine-tuning methods. The experiments show that existing watermarking approaches perform well in universality and transmissibility but are vulnerable in realistic attack scenarios. To address this, the authors propose DeAttack, a unified watermark removal framework based on image degradation and restoration, which effectively removes watermarks without harming model performance. This work exposes the vulnerability of current watermarking techniques and provides a baseline for future research on robust traceability.

**Strengths:**

Unified threat model and evaluation framework:
The paper is the first to establish a unified threat model for dataset watermarking in diffusion models and proposes a three-dimensional evaluation framework (universality, transmissibility, robustness). It clearly defines the essential requirements that watermarking should meet under realistic scenarios and provides a theoretical foundation for future research.

Comprehensive benchmark construction:
Based on multiple datasets and fine-tuning methods, the authors systematically evaluate four representative watermarking techniques (DIAGNOSIS, DiffusionShield, WatermarkDM, SIREN), constructing a reproducible and extensible benchmark that can serve as a unified experimental platform for future comparisons in both academia and industry.

DeAttack framework revealing vulnerabilities:
The proposed DeAttack framework unifies multiple degradation and restoration strategies, combining pixel-space and latent-space perturbations to expose how existing watermarking methods fail under realistic attacks. This unified removal pipeline provides an important baseline for future work on developing more robust watermarking or defense mechanisms.

Practical significance of transmissibility analysis:
The paper’s treatment of transmissibility is particularly meaningful. Unlike traditional robustness analysis, the authors define transmissibility as the ability of a watermark to be learned and exhibited by the model even when only part of the training data is watermarked. This highlights the potential propagation of watermark signals from partial data subsets to the overall model, reflecting real-world risks such as partial data poisoning or localized watermark injection. It provides a practically relevant dimension for evaluating dataset watermarking effectiveness and security.

**Weaknesses:**

Limited methodological novelty:
Although DeAttack is presented as a unified framework for watermark removal, its technical innovation is limited. The framework largely combines existing degradation operations (blur, JPEG compression, additive noise) with common restoration models (VAE, diffusion, SwinIR) without introducing new algorithmic mechanisms or theoretical insights. The “representation-space noise” component remains conceptual and is not implemented in experiments. Overall, DeAttack is more of a systematic integration of prior methods than a fundamentally new attack or optimization strategy, resulting in limited methodological originality.

Unspecified watermark strength settings:
The paper states that multiple watermarking methods are compared under unified conditions but does not specify the exact watermark strength (e.g., perturbation amplitude, embedding weight, or visibility level). Since watermark strength directly affects detectability and robustness, this missing detail undermines reproducibility and fairness. The authors should explicitly report the watermark strength or parameter ranges used for each method to clarify the experimental consistency.

Insufficient Correlation Between Protection Ratio and Performance:
The detection performance across different watermark protection ratios within the same group of experiments, as shown in Figure 3, lacks a clear monotonic trend. This somewhat counter-intuitive result may raise questions regarding the consistency of the data. The authors are encouraged to include a brief analysis in the main text discussing whether this “low correlation” arises from early performance saturation of certain methods at low ratios, or from factors such as random sampling and model initialization. In addition, it may be beneficial to conduct further experiments using finer, exponentially spaced ratio intervals (e.g., 5%, 10%, 20%, 40%, 80%) to more clearly reveal the relationship between protection ratio and detection performance, thereby improving the interpretability and credibility of the results.

Insufficient and unclear data presentation:
The experimental presentation lacks completeness and clarity, which reduces readability and persuasiveness. Key tables (e.g., Table 3 and Table 4) omit baseline results on clean samples, making it difficult to quantify the effect of different degradation or restoration operations. Although the color scheme is explained, the use of mixed red/blue highlighting across multiple subtasks hinders cross-group comparison and makes the tables hard to interpret. In visual figures (e.g., Figure 3), only degraded or watermark-removed samples are shown without the original reference images, preventing intuitive visual comparison. Moreover, Figure 4 contains a spelling error (“Gaussian”). Overall, the presentation of results is incomplete and visually confusing. Including clean baselines, original reference images, and clearer visual design would significantly enhance interpretability.

**Questions:**

See Weaknesses.

---

### Official Review · Reviewer_N42p · 2025-10-31

**Soundness:** 1
**Presentation:** 2
**Contribution:** 1
**Rating:** 2
**Confidence:** 4

**Summary:**

This paper addresses the problem of tracing unauthorized fine-tuning of diffusion models by embedding imperceptible watermarks into training datasets. The authors claim to propose a unified evaluation framework for dataset watermarking methods, introducing three dimensions, i.e., "Universality," "Transmissibility," and "Robustness", to assess their performance. They further conduct experiments across different fine-tuning settings on 4 existing methods on Stable Diffusion v1.4. Finally, the paper presents a so-called "practical" watermark removal method that is said to expose vulnerabilities in existing approaches.

**Strengths:**

- The topic of this paper, i.e., tracing unauthorized fine-tuning or dataset usage in diffusion models, is important and relevant.

- The overall evaluation design appears reasonable, though this is mainly because it adheres closely to what previous works have already done.

**Weaknesses:**

- The claimed "unified evaluation framework" lacks sufficient detail and support. The authors repeatedly claim to evaluate all watermarking methods "under a unified threat model" and "in a fair manner", yet provide no experimental protocol or hyperparameter details on how the baselines are trained and evaluated. As far as I know, the baselines themselves are fundamentally different in nature. DIAGNOSIS and SIREN use a hypothesis testing mechanism (binary or one-class classification results on an image distribution) to determine model infringement, while WatermarkDM and DiffusionShield decode multibit watermarks and measure bit accuracy. These paradigms are not directly comparable, yet the paper does not explain how they were put under the same evaluation framework, nor how the "ACC" metric is computed. This raises strong concerns about the fairness and validity of the comparisons.
- The reliability and validity of the experimental protocal and experimental results are highly questionable. To be honest, I am shocked by some results reported in the paper, as many findings are drastically inconsistent with the well-established consensus in the community:
  - According to the evaluations in DIAGNOSIS, SIREN, and other literature [8, 9], WatermarkDM (which reuses the same decoder from Yu et al.) should perform close to random guessing (around 50% watermark bit accuracy) in the fine-tuning setting due to the limited learnability of image watermarks. In contrast, in my understanding, SIREN is explicitly designed for diffusion model dataset tracing. It introduces a learnability loss to address the unlearnable nature of prior watermarking methods. The original SIREN paper also reports to achieve nearly 100% detection accuracy across multiple datasets (even under FPR = $10^{-9}$). The original SIREN paper also evaluated settings with partial clean data and under certain attack conditions, still obtaining strong results. This makes the claim in the current paper that "SIREN exhibits the lowest performance, with a detection accuracy of approximately 50% across all datasets and fine-tuning methods, which is equiv-alent to random prediction" highly inconsistent and suspicious. More concretely,
    1.  If SIREN and DIAGNOSIS employ hypothesis testing, then random guess levels should correspond to the false positive rate (FPR, or alpha in hypothesis testing) rather than 50%. This suggests that the evaluation may not be conducted following the original configuration.
    2. Even more concerning, the authors' explanation for SIREN's failure "it was tailored for simple datasets such as CIFAR" (Line 303-305) is factually incorrect. The CIFAR dataset family were never used or even mentioned in SIREN’s original paper, and SIREN was evaluated exactly on datasets used in this paper (Pokémon, CelebA-HQ, WikiArt). This factual error largely weakens the credibility of the paper.
    3. From a common-sense perspective, it is also rather hard for me to believe that a method specifically designed to address this problem and accepted by a top-tier venue like S&P 2025 would perform at the level of random guessing under its own intended setting. I'm curious what happened to SIREN.
- In Line 269, the authors state "To ensure a fair comparison of each watermarking method’s ability to detect watermarks from generated images, we calibrate the watermark embedding strength across all methods, thereby guaranteeing that the fine-tuned generation results based on each watermarked dataset exhibit comparable performance in terms of FID and CLIP-T metrics." I have two major concerns about this statement:
  - It is unclear how this calibration ensures fairness. No calibration criteria or adjustment details are disclosed.
  - The fairness and influence of this calibration are highly questionable. Some methods (e.g., WatermarkDM and SIREN) directly embed watermarks using a fixed encoder-decoder without any tunable "strength" parameter, unlike DIAGNOSIS or DiffusionShield that have tunable triggers or $\ell_p$ bounds. I wonder how the authors handled this. For encoder-based methods, they should have been used as-is, as tuning their internal components may unfairly advantage or disadvantage them.
  - In addition, Figure 5 in Appendix A.8 shows that DIAGNOSIS produces noticeably degraded images (severe edge distortions) than other baselines (e.g., SIREN and DiffusionShield), which indicates that the calibration did not achieve "comparable quality" as claimed. Therefore, the statement of a "fair comparison" is not supported by any verifiable evidence.
- Under the proposed attack, the FID score degrades dramatically (up to 270+), far worse than the clean setting (224.7 in Table 2), yet the authors offer no discussion or analysis. The proposed removal attack is also barely better than baselines, as none of them significantly reduce DiffusionShield’s detection accuracy. This directly contradicts their earlier claims in Lines 90–93 and 105–107 that "existing methods exhibit vulnerability to the proposed watermark removal approach". The evidence provided does not support such claims.
- The benchmark misses several important baselines, such as GenWatermark [1], FT-Shield [2], Entruth [3], and $z$-Watermarking [10]. For a paper claiming to establish a comprehensive benchmark, this omission is disappointing.
- The writing and organization could be significantly improved. Specifically,
  - Figure 2 is highly similar to Figure 2 in [4] in both layout and components, yet the caption does not cite or acknowledge the source. Furthermore, the pipeline shown in Figure 2 does not align with the described methods: DIAGNOSIS and SIREN are not bit-string-based encoding schemes but rely on hypothesis testing from classifier outputs.
  - Figure 1 seems not informative and purely decorative.
- Only LoRA, Textual Inversion, and DreamBooth are evaluated, while these methods are quite out of date for ICLR 2026. At least some personalization methods in 2024 should be evaluated ([5-7] to cite a few). Besides, all experiments are conducted on the Stable Diffusion 1.4 model, which also raises concerns on the scalability of the conclusions to more advanced models (e.g., FLUX).

Overall, the "contribution" of this paper is quite limited. As a benchmark paper, it fails to include several state-of-the-art baselines, lacks detailed disclosure of evaluation setups and fairness criteria, and provides little or no analysis explaining why certain methods succeed or fail. For instance, DiffusionShield maintains nearly 100% accuracy even under strong attacks, yet no discussion is provided; meanwhile, the analysis for SIREN contains factual errors and misinterpretations. The reported results are questionable (e.g., SIREN performing worse than expected while WatermarkDM performs unusually well). Aside from showing a few possibly erroneous experimental results and a marginally effective attack, this paper offers minimal insight to the community. I thus recommend rejection.

References:

[1]: Ma, Yihan, et al. "Generative watermarking against unauthorized subject-driven image synthesis." arXiv preprint arXiv:2306.07754 (2023).

[2]: Cui, Yingqian, et al. "Ft-shield: A watermark against unauthorized fine-tuning in text-to-image diffusion models." ACM SIGKDD Explorations Newsletter 26.2 (2025): 76-88.

[3]: Ren, Jie, et al. "EnTruth: Enhancing the Traceability of Unauthorized Dataset Usage in Text-to-image Diffusion Models with Minimal and Robust Alterations." arXiv preprint arXiv:2406.13933 (2024).

[4]: Zhao, Yunqing, et al. "A recipe for watermarking diffusion models." arXiv preprint arXiv:2303.10137 (2023).

[5]: Ding, Ganggui, et al. "Freecustom: Tuning-free customized image generation for multi-concept composition." Proceedings of the IEEE/CVF Conference on Computer Vision and Pattern Recognition. 2024.

[6]: Pang, Lianyu, et al. "Attndreambooth: Towards text-aligned personalized text-to-image generation." Advances in Neural Information Processing Systems 37 (2024): 39869-39900.

[7]: Wei, Fanyue, et al. "Powerful and Flexible: Personalized Text-to-Image Generation via Reinforcement Learning." European Conference on Computer Vision (ECCV), 2024.

[8]: Dubiński, Jan, et al. "Are Watermarks For Diffusion Models Radioactive?." The 1st Workshop on GenAI Watermarking.

[9]: Liu, Zhenguang, et al. "Harnessing Frequency Spectrum Insights for Image Copyright Protection Against Diffusion Models." Proceedings of the Computer Vision and Pattern Recognition Conference. 2025.

[10]: Huang, Junqiang, et al. "Disentangled Style Domain for Implicit $z$-Watermark Towards Copyright Protection." Advances in Neural Information Processing Systems 37 (2024): 55810-55830.

**Questions:**

- How is the "ACC" metric calculated in your paper? Why are you adopting DIAGNOSIS and SIREN to this "ACC" metric while they are originally designed under the hypothesis testing regime?

- Why does SIREN perform so poorly while WatermarkDM perform so well, which seems contradictory to existing findings? I have a bold guess: could it be that the two were accidentally swapped?

- In my understanding, FT-Shield was specifically designed for fine-tuning tracability while DiffusionShield was not, although they come from the same group of authors. Why was DiffusionShield chosen for evaluation while FT-Shield was not? To me, FT-Shield is definetely more fitted for the intended setting.

- What do you mean by "The poor performance of SIREN can be attributed to its original design, which is tailored for simple datasets such as CIFAR, and its limited generalization capability for more complex datasets."?

- Could you use images with higher resolution in the figures of this paper? Especially for Figure 5, where all Pokémon images are so blurry and details cannot be sufficiently compared (at least for me).

---

### Official Review · Reviewer_S6mT · 2025-11-04

**Soundness:** 2
**Presentation:** 2
**Contribution:** 2
**Rating:** 4
**Confidence:** 4

**Summary:**

See the Strengths and Weaknesses.

**Strengths:**

- Proposes a standardized evaluation framework (Universality/Transmissibility/Robustness) addressing critical lack of comparable metrics in dataset watermarking research.
- Through comprehensive experiments, demonstrates that while existing methods resist common distortions, they remain highly vulnerable to watermark removal attacks - exposing a crucial real-world weakness.
- Introduces DeAttack, a practical watermark removal framework that successfully breaks current methods, setting higher robustness standards for future research.

**Weaknesses:**

- Insufficient Motivation and Context: The paper does not clearly convey the serious security and ethical concerns associated with fine-tuning diffusion models, making it difficult for readers to grasp the significance of the problem.
- Lack of Detail on Threat Model Definitions: The discussion of varying definitions of threat models in existing dataset watermarking approaches is vague. More elaboration is needed to clarify how these differences motivate the current research.
- Inadequate Review of Related Work: The claim that adversarial attacks for watermark removal are underexplored is not well supported, given the existence of extensive prior studies. The paper lacks a detailed analysis of the limitations of previous works and does not sufficiently differentiate its proposed approach.
- Ambiguity in Terminology and Scope: The use of the term "threat model" is potentially misleading, as it is typically associated with adversarial attack scenarios. Additionally, it is unclear whether the three identified challenges and evaluation dimensions are comprehensive, or if other important challenges have been overlooked.
- Problematic Citation Style: The way references are cited in the main text is confusing and inconsistent, making it difficult for readers to follow and verify the sources.
- Superficial Experimental Analysis: The analysis of experimental results is brief and lacks depth. Given that comprehensive evaluation is a main contribution of the paper, the limited analysis fails to provide meaningful insights.
- Limited Technical Detail: The technical description of DeAttack is insufficiently detailed, and the key technical contributions are not clearly articulated

**Questions:**

- In lines 43-45, from Figure 1, I cannot get the serious security and ethical concerns raised by the fine-tuning diffusion models.
- In lines 51-53, the author claims “However, existing approaches exhibit varying definitions of threat models for dataset watermarking, which complicates the uniform evaluation of their performance in practical applications.” Can you elaborate more on the varying definitions of existing approaches? This is the core motivation of this research, I think more details are needed.
- In lines 87-89, the authors claim that adversarial attacks for watermark removal have not been sufficiently addressed in prior research. However, there are extensive studies investigating watermark removal (e.g., [1,2]). I think the authors should provide a more detailed motivation regarding the limitations of previous works, as well as clarify how the approach proposed in this paper differs from existing methods.
- In Section 3.1, the authors propose a universal threat model. However, I have the following concerns: First, why do the authors use the term 'threat model'? This term is typically used in the context of adversarial attacks to refer to the attack model. Second, the authors identify three challenges corresponding to the three evaluation dimensions discussed later. I would like to know whether these three challenges are comprehensive, or if there are other potential challenges that should be considered.
- The way the authors cite references in the main text appears to be problematic (for example, WatermarkDM Zhao et al. (2023), stable diffusion 1.4 (SD1.4) Rombach et al. (2022) in lines 208–215). This makes it quite difficult for readers to follow the text.
- In sections 4.3 and 4.4, the authors' analysis of the experimental results is rather brief. Since one of the main contributions of this paper is a comprehensive evaluation, insufficient analysis often fails to provide insightful perspectives for the readers.
- The authors’ description of DeAttack’s technical details is rather brief. What are the key technical contributions of DeAttack?

[1] Sun, Ruizhou, Yukun Su, and Qingyao Wu. "Denet: Disentangled embedding network for visible watermark removal." Proceedings of the AAAI Conference on Artificial Intelligence. Vol. 37. No. 2. 2023.

[2] Bui, Tu, Shruti Agarwal, and John Collomosse. "TrustMark: Robust Watermarking and Watermark Removal for Arbitrary Resolution Images." Proceedings of the IEEE/CVF International Conference on Computer Vision. 2025.

---

### Note · Authors · 2025-11-13

I have read and agree with the venue's withdrawal policy on behalf of myself and my co-authors.